# Advances of Hydrogel Therapy in Periodontal Regeneration—A Materials Perspective Review

**DOI:** 10.3390/gels8100624

**Published:** 2022-09-30

**Authors:** Maoxue Li, Jiaxi Lv, Yi Yang, Guoping Cheng, Shujuan Guo, Chengcheng Liu, Yi Ding

**Affiliations:** State Key Laboratory of Oral Diseases, Department of Periodontics, West China School & Hospital of Stomatology, Sichuan University, Chengdu 610041, China

**Keywords:** hydrogel, periodontal tissue regeneration, tissue engineering, periodontitis, delivery system, scaffold material

## Abstract

Hydrogel, a functional polymer material, has emerged as a promising technology for therapies for periodontal diseases. It has the potential to mimic the extracellular matrix and provide suitable attachment sites and growth environments for periodontal cells, with high biocompatibility, water retention, and slow release. In this paper, we have summarized the main components of hydrogel in periodontal tissue regeneration and have discussed the primary construction strategies of hydrogels as a reference for future work. Hydrogels provide an ideal microenvironment for cells and play a significant role in periodontal tissue engineering. The development of intelligent and multifunctional hydrogels for periodontal tissue regeneration is essential for future research.

## 1. Introduction

Periodontitis is a chronic, destructive inflammation characterized by microbial infection and accelerated loss of alveolar bone, ultimately resulting in the loss of teeth [1]. As the world ages, periodontitis has become one of the major oral diseases, affecting a significant number of people around the world. Epidemiological evidence shows that approximately 20–50% of the global population suffers from periodontal-related diseases, and approximately 10% of the global population is affected by severe periodontitis [2]. There is a pressing need to address the challenges of periodontitis and bone loss in older populations [3]. Furthermore, it is now well accepted that periodontal disease is strongly associated with systemic diseases such as diabetes, cardiovascular disease, Alzheimer’s disease, and other inflammatory comorbidities [4]. Whether in developed or developing countries, periodontitis imposes severe medical burdens on the population [5]. Managing periodontal diseases and promoting periodontal tissue regeneration are essential for oral and general health.

Periodontal tissue is the functional system surrounding teeth and has a complex hierarchical structure comprising hard and soft tissue together as a whole. Periodontal tissue regeneration involves the reconstitution of periodontal ligament (PDL) and alveolar bone around the teeth and cementum [6]. The ideal goal of periodontal treatment is to achieve good regeneration of the damaged periodontal tissue. Conventional periodontal therapies such as mechanical debridement and flap surgery are mainly aimed at removing plaque and pathological granulation tissue to prevent the progression of inflammation and the further destruction of periodontal tissue [7]. Reconstruction of the morphology and function of the damaged periodontal tissue is the ideal objective of periodontal regeneration treatment but remains a major challenge [8]. Guided tissue regeneration and bone grafts are currently the clinical approaches to periodontal tissue regeneration, but these techniques have shown limitations in indications such as intraosseous defects and class II fissure defects. The high technical sensitivity also limits their regenerative effect [9]. Periodontal tissue engineering has emerged as a promising technology to address periodontal diseases and is a technique that uses a combination of stem cells, biological scaffold material implanted in the body, and growth factors to promote periodontal tissue regeneration [10]. In recent years, a large range of scaffold materials has been designed to promote alveolar bone formation and are currently one of the main ways to restore periodontal tissue that has been damaged by inflammation. Ideal biomaterials can effectively recruit regeneration-related functional cells, promote their proliferation and differentiation, and lead to the formation of new periodontal tissues [11]. However, the main challenge in periodontal regeneration therapy currently arises from the simultaneous or sequential repair of the morphology and function of PDL, alveolar bone, and cementum [12]. Traditional two-dimensional biomaterials, such as GTR barriers, provide adhesion sites and prevent soft tissue from growing into bone defects, but biological stimulation for functional cells is limited.

Hydrogels are three-dimensional water-swollen polymeric materials with superior biocompatibility, mechanical strength, and accessibility that have been widely used in biomedical applications such as cell culture [13], drug delivery [14], and tissue engineering [15]. In tissue engineering, biomaterials provide a three-dimensional scaffold for cell adhesion, proliferation, and differentiation. The scaffold should be a porous, three-dimensional, network-like structure providing cells with the necessary space to deposit their extracellular matrix while exchanging cellular substances with the surrounding environment. Due to their distinctive three-dimensional mesh structure, high porosity, superior hydrophilicity and viscoelasticity, and controllable compositions, hydrogels can mimic the microenvironment of the extracellular matrix, which is favorable for cell attachment, proliferation, and differentiation. Through combination with drugs [16], stem cells [17], or growth factors [18], hydrogels show significant potential in periodontal regeneration and have gained a considerable amount of attention in recent years. Periodontal tissue regeneration is a complex and sophisticated process. Hydrogels have been widely applied as scaffolds for regenerative medicine and as a sustained-release system in periodontal tissue engineering. Current research has noted that the composition and structure of hydrogels have a significant impact on periodontal tissue regeneration. However, there is no paper summarizing these impacts which may pave the way for researchers to develop appropriate hydrogel designs in periodontal tissue engineering. Based on the above, this paper reviews the applications of hydrogels in periodontal tissue regeneration research and provides discussions and prospects about their future designs, with the objective of making a valuable contribution to successful periodontal tissue regeneration.

## 2. Components of Hydrogel in Periodontal Tissue Regeneration

The fundamental components of hydrogels determine the properties and function of the material. Additionally, hydrogels can encapsulate bioactive substances to provide the hydrogels with antibacterial, anti-inflammatory, osteogenetic, and osteoimmunology capabilities and improve the regenerative effect of periodontal tissue as needed. This section focuses on the main components of hydrogels and the substances carried on them, as summarized in Figure 1.

### 2.1. The Fundamental Components of Hydrogels in Periodontal Tissue Regeneration

#### 2.1.1. Natural Polymers

The fundamental components of hydrogels are mainly categorized into natural and synthetic polymers. Natural polymer materials are directly derived from natural resources. Since the ECM consists primarily of polysaccharides, glycosaminoglycans, and a variety of proteins, natural polymers generally have excellent biocompatibility and biodegradability, and most of these polymers are water-soluble. The hydrophilic surface facilitates cell adhesion, proliferation, and differentiation. However, the mechanical strength and stability of natural polymers are not as high as those of synthetic hydrogels, which also limits their application to some extent [19].
Chitosan

Chitosan (CS) is a natural polysaccharide with a chemical structure and biological properties similar to those of glycosaminoglycan, a major component of the extracellular matrix [20]. With good biocompatibility, hemostatic properties [21], much higher adhesiveness [22] and antimicrobial activity [23], CS can potentially be used for the synthesis of various gels in periodontal therapy. Yan, X. Z. et al. [24] reported that enzymatically solidified chitosan hydrogels with or without cell loading showed great potential in periodontal regeneration in terms of functional ligament length. Histological analysis demonstrated that after four weeks of implantation, chitosan hydrogels were largely degraded, without causing any adverse reactions in the surrounding tissue. Furthermore, crosslinking modification of chitosan can form hydrogels with certain swelling properties. Ji, Q. X. et al. [25] developed CS-based thermosensitive hydrogels composed of CS, quaternized CS, and beta-glycerophosphate which showed stronger antibacterial activity toward periodontal pathogens. In addition, CS is the only natural polycationic alkaline polysaccharide possessing a large number of free amino groups, which could enable electrostatic interactions and hydrogen bonding to form hydrogels without using any additional agents [26] and could be modified by introducing other groups (such as carboxyl) to control drug release behaviors [27]. However, CS-only hydrogels have brittle properties and poor mechanical strength and show low solubility at a physiological pH of 7.4, which limits their application [28]. Researchers have reported that drug-loaded chitosan gels and CS gels with the addition of synthetic materials have better mechanical and biomedical properties than chitosan-onlygels [26,29]. Zhang, Y. et al. [30] constructed a poly(ethylene glycol) (PEG) and CS composite gel by utilizing PEG to enhance the high mechanical strength of the hydrogel and encapsulate acetylsalicylic acid (ASA) via electrostatic interactions. The composite gel’s sustained release of ASA for over two weeks promoted the proliferation and osteogenic differentiation of periodontal ligament stem cells (PDLCs) and enhanced bone regeneration in a mouse calvarial bone defect model.
Sodium alginate

Sodium alginate is a natural, high-productivity polyanionic polysaccharide derived primarily from brown seaweed and bacteria (nitrogen-fixing bacteria and *Pseudomonas*) [31]. Sodium alginate-based hydrogels have been extensively applied in wound healing [32], drug delivery [33], and cell transplantation [34] because of their high water content and favorable biocompatibility. Nevertheless, sodium alginate-based hydrogels have several deficiencies as they are difficult to degrade under physiological conditions in mammals because of the lack of enzymes to degrade, which is undesirable for periodontal regeneration. Recent studies reported that oxidation of alginate by periodate could increase its biodegradation rate and improve its biological safety in the long term [35]. Furthermore, cells prefer to adhere to neutral or cationic interfaces. Sodium alginate-based hydrogels have poor cell adhesion in mammals as a result of the formation of a hydrated surface layer that lacks substances useful for cell growth and adhesion [36]. To address these problems, researchers modified sodium alginate-based hydrogels by incorporating other components or improving crosslinking methods in recent studies. Xiong, X. et al. [37] developed negatively charged alginate and chondroitin sulfate microsphere hydrogels (nCACSMH) for cell delivery. The nCACSMH enhanced the attachment and proliferation of human umbilical vein endothelial cells (HUVECs) and upregulated angiogenesis-related gene expression in endothelial cells. Calcium ions, zinc ions, and other divalent metallic ions are capable of reacting with alginate to form hydrogels [33,38]. However, alginate-based hydrogels exhibit poor stability in the gel structure in the traditional physical crosslinking approach [39].
Hyaluronic acid

Hyaluronic acid (HA) is a natural linear glycosaminoglycan abundantly found in the human body that consists of repeating units of N-acetyl-d-glucosamine and d-glucuronide; the highest concentrations of this compound are in the eyes and joints [40]. Hyaluronic acid (typically existing in the form of polymers weighing over 106 daltons) can be cleaved by hyaluronidase, and its biological function is related to its molecular weight [41]. Studies have reported that high-molecular-weight HA has anti-inflammatory and immunosuppressive effects, as well as promoting migration in gingival fibroblast cells, which may have beneficial effects on periodontal inflammation [42]. Furthermore, as with most natural polymer hydrogels, hyaluronic acid hydrogels have low cell recognition and adhesion rates due to a lack of cell adhesion sites and anionic properties [43]. To advance their application in periodontal tissue engineering, hyaluronic acid hydrogels must be compounded with other materials to enhance their mechanical properties and adhesion [31,44]. Ansari, S. et al. [31] designed a 3D alginate/HA hydrogel encapsulating PDLSCs that upregulated gene expression related to chondrogenesis (col II, aggrecan, and sox-9) in vitro and showed a greater positive expression of chondrogenic specific protein markers in vivo. To restore the anatomy and functionality of lost periodontal tissue, Babo, P. S. et al. [45] prepared an injectable hydrogel consisting of methacrylate hyaluronic acid (me-HA) and platelet lysate (PL) to release growth factor proteins in situ. Adding PL to me-HA hydrogels improved their viscoelastic properties (*p* < 0.02) and resilience to degradation by hyaluronidase while inhibiting bacterial growth. The versatile hydrogels provided adequate space and stability for cell adhesion and proliferation, showing great potential in periodontal therapy.
Collagen

Collagen (Col) is an essential component of the extracellular matrix, with a variety of cell signaling binding sites and excellent biocompatibility, and is widely used in the development of periodontal tissue engineering scaffolds [18,46,47]. Collagen can support the adhesion, growth, proliferation, and directed differentiation of functional cells associated with periodontal regeneration [48]. However, collagen-based hydrogels have poor mechanical properties and stability, and terrestrial animal-derived collagen may cause immune reactions. Therefore, some researchers are exploring new collagen sources for biomaterials. Zhou, T. et al. [49] found that the comprehensive properties of blended hydrogels composed of polyvinyl alcohol (PVA) and fish Col can be regulated by controlling the content of PVA and Col to be suitable for guided tissue regeneration. As the proportion of PVA in the blended hydrogel increased, the mechanical properties of the hydrogel increased but were detrimental to human gingival fibroblast (HGF) growth. In contrast, an increase in collagen content enhanced the surface porosity of the hydrogels and their biocompatibility with PDLCs. The PVA/Col (50:50) blended hydrogel exhibited the highest cell proliferation rate for HPDLCs with spread cell morphology.
Others

Other natural polymers, such as gelatin [50,51], chondroitin sulfate [37], and silk proteins [52], are also used in tissue engineering. These have been shown to have excellent biocompatibility, degradability, and cytocompatibility.

#### 2.1.2. Synthetic Polymers

Synthetic polymers are prepared through chemical reactions. Common synthetic compounds, such as polyethylene glycol (PEG) [53], PVA [49], and poly(lactic-co-glycolic acid) (PLGA) [54], are generally accessible and can be tailored to achieve excellent mechanical properties and hydrogel stability but lack inherent bioactivity. However, the biocompatibility and degradability of synthetic polymer-based hydrogels are not as good as those of natural polymer-based hydrogels [30]. The following section discusses several types of synthetic polymer-based hydrogels.
PEG

PEG is a hydrophilic and biocompatible synthetic polyether that is a promising hydrophilic biomaterial for periodontal regeneration and is well known for its flexibility, biocompatibility, and hydrophilicity [13]. Since being approved by the FDA, PEG has been widely applied in biomedical research [55]. The well-defined chemistries of PEG enable the precise insertion of cell-responsive and bioactive components into a hydrogel [56,57]. To investigate the specific response of PDLCs to ECM biophysical and biochemical cues, Fraser, D. et al. [58] used PEG hydrogels with peptides to enable MMP-mediated matrix degradation and/or PDLC integrin-matrix binding to mimic the ECM in periodontitis. Additionally, Zhang, Y. et al. [59] fabricated a tetra-PEG hydrogel for the in situ encapsulation of aspirin, ensuring sustained release, anti-inflammatory, and osteoinductive properties to composite hydrogels. In vitro experiments indicated that aspirin-loaded tetra-PEG hydrogels facilitate the proliferation and osteogenic differentiation of human periodontal cells, and in vivo results demonstrated that hydrogels could remarkably promote bone regeneration.
Gelatin methacryloyl (GelMA)

GelMA is a photosensitive hydrogel material with excellent biocompatibility and the capacity to enable cell encapsulation that is manufactured from methacrylic anhydride (MA) and gelatin (gelatin) [60]. Generally, GelMA hydrogels are cured by UV or visible light and have recently been developed to mimick the 3D cell microenvironment [61]. A study [17] demonstrated that GelMA hydrogels could provide a physical microenvironment for PDLCs to adhere and grow, ensuring, as closely as possible, no loss of PDLCs during the transplantation process and stable performance of their biological functions. Ma, Y. et al. [62] designed a composite gel composed of GelMA and poly (ethylene glycol) dimethacrylate (PEGDA) encapsulating PDLCs using 3D printing technology. The physical and biological properties of the composite hydrogels differed by adjusting the ratio of GelMA and PEGDA. In this study, cell proliferation, spreading, and osteogenic differentiation increased as the GelMA volume ratio increased. An optimized composition (the 4/1 GelMA/PEGDA hydrogel) was selected to treat periodontal defects, and it showed a significant impact on promoting the regeneration of functional tissue.
Others

In addition, other synthetic polymers also show significant potential in periodontal therapy. PVA is a water-soluble polymer hydrolyzed from polyvinyl acetate with good biocompatibility, a high modulus of elasticity, and easily adjustable physical properties. The incorporation of PVA and natural polymers significantly compensated for the poor mechanical properties of natural polymer-based hydrogels and remained favorable for cytocompatibility and bioactivity [63,64]. PLGA is a non-toxic, degradable polymeric organic compound made by the polymerization of lactic acid and hydroxyacetic acid which has high biocompatibility and capsule- and film-forming properties as a carrier for cell implantation [65].

### 2.2. The Multiple Components of Hydrogels in Periodontal Tissue Regeneration

Periodontal tissue has a complex structural and compositional composition, including soft and hard tissue. Moreover, periodontitis is a chronic inflammatory disease caused by bacterial infection with an intricate pathology: pathogenic microorganisms cause inflammation of the gingiva, which overactivated the immune response and results in tissue destruction [66]. As the disease progresses, the host and microorganisms release a variety of proteases and proinflammatory cytokines that stimulate bone resorption [67]. Multiple systems, signaling pathways, and molecules are involved in the regeneration of periodontal tissue, including the skeletal, immune, and circulatory systems [68]. Materials with single components and structured structures are generally difficult to regenerate effectively. To simultaneously fulfill the mechanical and cell-compatible requirements of biomedical applications, multiple components must be arranged chronologically when designing material systems. As needed, hydrogel systems can carry chemicals, growth factors, nanoparticles, exosomes, and stem cells in their polymeric structure, prevent their dissolution, and provide a slow and controlled release to achieve regenerative therapy.
Antibacterial agents

Hydrogels with antibiotic loading may exhibit enhanced antibacterial properties. Periocline (Perio) is a generally recommended adjunct to scaling and root planning for adult periodontitis in clinical practice and is essentially a 2.1% minocycline gel. However, there are some adverse effects of commercial Perio, including photosensitivity and permanent discoloration of developing teeth [69]. Metronidazole (MTZ) can effectively kill anaerobic bacteria and inhibit bacterial growth to control inflammation. Since MTZ is water-soluble, topical application is easily diluted by saliva and gingival sulcus, resulting in inadequate release. Dong, Z et al. [70] prepared metronidazole microcapsules (CS@MTZ) using the polysaccharide coprecipitation method, while PVA@CS@MTZ hydrogels were prepared by substituting CS@MTZ microcapsules with ions. The safety, biocompatibility, and antibacterial effect of the PVA@CS@MTZ gel were demonstrated through in vitro and in vivo experiments, suggesting that it is a promising therapeutic agent for periodontitis.
Cytokines

Cytokines are involved in regulating cell proliferation, differentiation, immune response, and intercellular interactions in periodontal hard tissue regeneration and mineralization [71]. Growth factors such as bone morphogenetic proteins (BMPs) [56] and vascular endothelial growth factor (VEGF) [72] have been demonstrated to induce matrix mineralization and participate in bone formation and bone reconstruction. The secretion of proinflammatory cytokines and macrophages polarized into the proinflammatory phenotype would have negative effects on the osteogenesis of MSCs for periodontal regeneration [68]. Some cytokines could regulate the progression of inflammation and macrophage-polarization. For example, IL-4 and FGF-2 play significant roles in anti-inflammation by polarizing macrophages toward the anti-inflammation phenotype, mitigating the foreign body reaction of hydrogels, and inhabiting osteoclast function, resulting in acceleration of tissue healing and successful periodontal regeneration [50,73]. Researchers [74,75] found that increased gel stiffness supported mesenchymal stem cell proliferation and osteogenic differentiation, whereas stiff gel is more likely to polarize macrophages toward the pro-inflammatory M1 phenotype and increase inflammatory factor release, which is detrimental to bone formation. Based on this, He, X. T et al. [50] introduced interleukin (IL)-4 and stromal cell-derived factor (SDF)-1α into transglutaminase crosslinked gelatins (TG-gels) to regulate macrophage polarization and promote endogenous stem cell recruitment. The results demonstrated that TG-gels containing both IL-4 and SDF-1a could induce stem cell homing, modulate cell differentiation, and indeed induce the regrowth of periodontal tissue. SDF-1α plays a pivotal role in BMMSC transplantation therapy and tissue regeneration by recruiting circulating or residing stem cells to the injury site, regulating macrophage polarization, and facilitating osteogenesis in vivo [76]. Tan, J et al. [77] prepared a supramolecular NapFFY hydrogel that encapsulates both SDF-1 and BMP-2, and their study demonstrated that an SDF-1/BMP-2 hydrogel could promote periodontal bone regeneration.
Mesenchymal stem cells and exosomes

MSC-based tissue engineering combined with injectable hydrogels has been extensively investigated for periodontal regeneration [62,78,79,80]. The common MSCs applied in periodontal engineering include BMSCs [76], stem cells from human exfoliated deciduous teeth (SHEDs) [78], dental follicle cells [79] and PDLCs [62]. PDLCs were demonstrated to possess multiple differentiation potentials and high proliferative ability and could be differentiated into progenitor cells making up cement, PDL, and alveolar bone, being considered the most suitable MSCs for periodontal regeneration [62].

More recently, cell-free tissue engineering has been significantly developed. Extracellular vesicles (EVs) are indispensable paracrine mediators and have a significant therapeutic effect on periodontal regeneration without any other toxic effect [32,81]. Studies suggest that exosomes play a significant role in tissue regeneration by regulating the immune microenvironment, promoting angiogenesis, balancing bone metabolism, and participating in mineralization [79]. However, it is difficult to achieve therapeutic concentrations of EVs alone for topical or systematic application. Hydrogels are suitable carriers for loading exosomes [32,81]. Huang, C. C et al. [82] developed a 3D encapsulating and tethering photo-crosslinked alginate hydrogel system to prolong EV delivery in vivo and maintain the structural and functional integrity of EVs simultaneously, which had a superior performance in bone regeneration.
Inorganic nanoparticles

Inorganic nanoparticles have attracted considerable attention in recent years as they could be used both as carriers for delivering drugs and as medicine, showing great potential and safety in the field of medical application [83]. The structural chains of hydrogels contain a significant number of reactive groups that can bond with inorganic nanoparticles [84]. Furthermore, inorganic nanoparticles such as mesoporous silica could serve as carriers for loading various drugs and bioactive substances [78] and have multiple effects such as antibacterial and osteogenic abilities [85]. In light of these findings, combining inorganic nanoparticles with hydrogels could optimize the mechanical properties and biological function of hydrogel materials to achieve the objective of promoting tissue regeneration [86]. Zeolitic imidazolate framework-8 is a porous crystalline material self-assembled by zinc ions and 2-methylimidazole ligands, with a large specific surface area, high porosity, easy synthesis, and controllable dimensions, and it has been applied in the treatment of periodontitis and bone regeneration [87,88]. Liu, Y et al. [89] developed a nano-injectable photosensitive GelMA composite hydrogel loaded with ZIF-8 for the treatment of periodontitis. The ZIF-8/GelMA hydrogel could release Zn^2+^ continuously and had good cytocompatibility. GelMA-Z effectively upregulated the expression of osteogenic genes and proteins, increased alkaline phosphatase activity, promoted extracellular matrix mineralization in rat bone mesenchymal stem cells, and showed significant antibacterial effects against *Porphyromonas gingivalis.* In vivo, GelMA-Z reduced bacterial load, decreased inflammation, and promoted alveolar bone regeneration in a rat model, thus comprising a promising therapy for periodontitis.
Natural compounds

Currently, natural compounds derived from herbs have also received substantial attention in medicine. Various natural substances from herbs possess a range of properties, such as anti-inflammatory, antibacterial, antioxidant, and growth factor-promoting properties [26,84]. Puerarin (PUE), a natural flavonoid, exhibits anti-inflammatory, antibacterial, and antioxidant properties [90]. Ferulic acid (FA) is a phenolic compound with excellent antioxidant activity. Ou, Q et al. [91] incorporated PUE and FA into polydopamine (PDA) nanoparticles (NPs) to prepare polyethylene glycol diacrylate (PEG-DA) composite hydrogels, exhibiting excellent mechanical and antioxidant properties. Ginsenoside Rg1, a component derived from the natural extract of ginseng, enhances the proliferation and osteogenic differentiation of hPDLSCs, with a favorable anti-inflammatory ability [92]. However, ginsenoside Rg1 can be hydrolyzed in a short time by matrix metalloproteinase. Guo, H et al. [93] developed an injectable self-healing hydrogel that achieved more than 6 days of sustainable release of ginsenoside Rg1, which might better facilitate periodontal regeneration in periodontitis.

## 3. Strategies of Hydrogels in Periodontal Tissue Regeneration

Hydrogels provide a survival space for cells to exchange nutrients and gases, regulating cell morphology and function. While hydrogels have many advantages, due to their poor mechanical properties, adjustments to the hydrogel components, network structure, gelation process, and crosslinking are often needed to achieve hydrogels with appropriate mechanical strength to improve tissue regeneration [86,94]. There are two main methods of hydrogel preparation: chemical crosslinking and physical cross-linking. Physical crosslinking refers to connections through ionic interactions, electrostatic interactions, hydrophobic interactions, crystallization, and hydrogen bonding [70,95], while chemical crosslinking reactions include Michael’s addition reaction, Schiff’s base reaction, the Diels-Alder cycloaddition reaction, and free radical polymerization [35,96,97].

The effect of biomaterials on tissue regeneration is mainly enforced through the interaction of cells with the biomaterial surface. Integrins are heterodimeric receptors on cell membranes that are involved in the regulation of biological behaviors such as cell morphology, migration, proliferation, and differentiation by binding to adhesion proteins on the surface of biomaterials [98]. The chemical composition, mechanical properties, hydrophilicity, and morphology of biomaterials are key factors regulating the control of cellular behaviors by the corresponding materials [99]. As a result, designing and processing the material by selecting the appropriate build-up is crucial to promote periodontal tissue regeneration. On the other hand, conventional three-dimensional hydrogels maintain a fixed shape without actively adapting to the changes occurring within the healing tissue. This leads to the development of four-dimensional hydrogels whose geometry changes with time or external stimulations [100]. Later, we will review the main current strategies for constructing hydrogels in periodontal tissue engineering.

### 3.1. Biomimetic Hydrogel

Biomimetic materials are biological materials that mimic the composition or structure of natural tissues. The extracellular matrix (ECM) is a 3D meshwork consisting of macromolecular substances (e.g., polysaccharides and proteins) secreted by cells. In the physiological environment, cells exist in a complex microenvironment characterized by both intercellular interactions and heterogeneous ECM. Cues from the ECM could regulate cellular functions, such as proliferation, apoptosis, migration, and differentiation, and affect biosynthesis [101]. As shown in Figure 2A, researchers found that cells cultured on 2D surfaces are flat and have a forced apical-basal polarity, which is unnatural for most mesenchymal cells. However, when embedded in a 3D ECM, cells can regain their physiological form and function [102]. Additionally, mesenchymal cells tend to adhere, proliferate, and differentiate into specific phenotypes when exposed to a matrix of similar tissue-level elasticity [103]. With advances in periodontal tissue engineering and materiobiology, scientists have gradually realized that the material does not simply provide a scaffold for cells and growth factors to attach to in vivo, but rather, the physicochemical properties of the material could affect the host response and thus tissue regeneration [8,99]. Mimicking the composition and structure of the ECM and providing appropriate biological stimulation to cells are the most attractive advantages of biomimetic hydrogels and show a favorable advantage in periodontal regeneration [104,105]. In light of these findings, utilizing ECM cues to control the activity of cells in vitro/in vivo and ultimately design biomaterials has excellent potential to enhance periodontal tissue regeneration [62].

On the other hand, periodontal tissue has a sophisticated composition and structure, in which periodontium, alveolar bone, and cementum form a functionally and structurally multi-layered integration known as the periodontal complex [12]. Once periodontitis progresses, the structural and functional integrity of the periodontal complex is disrupted. According to the characteristics of each hierarchical layer of the periodontal complex (cementum-periodontium-alveolar bone), researchers have designed hierarchical hydrogel scaffolds simulating the “sandwich” structure of the periodontal complex and combined hydrogels with specific drugs/bioactive factors to guide the directional differentiation of PDLSCs, leading to the ideal effect of periodontal tissue regeneration, especially in remodeling the periodontal ligament (PDL) [106,107]. Sowmya, S. et al. [108] developed a porous trilayered nanocomposite hydrogel scaffold similar to the structure of the cementum-periodontium-alveolar bone complex. The hierarchical hydrogel wrapped bioactive molecules in different layers of the scaffold illustrated in Figure 2B to induce tissue regeneration, and the sustained release of growth factors lasted up to 14 days. The cementum layer was composed of chitin-poly (lactic-co-glycolic acid) (PLGA), nanobioactive glass ceramic (nBGC) and cementum protein 1; the PDL layer was composed of chitin, PLGA and fibroblast growth factor 2;and the alveolar bone layer was composed of chitin, PLGA, nBGC and platelet-rich plasma derived growth factors. Both in vivo and ex vivo experiments also showed complete defect healing, favorable formation of new cementum, fibrous PDL, and alveolar bone with well-defined bony trabeculae, thus demonstrating great potential for periodontal complex regeneration.

### 3.2. Intelligent Hydrogels

Intelligent hydrogels, also called stimulation-responsive hydrogels, can be responsive to slight alterations to specific external stimuli. Depending on the specific stimulus perceived, smart hydrogels can be further divided into thermosensitive hydrogels, pH-sensitive hydrogels, photosensitive hydrogels, and other stimulation-responsive hydrogels.

Thermosensitive hydrogels are one of the hydrogels that have received the utmost attention and can react accordingly to external temperature alterations, performing crucial roles in drug delivery [109], cell encapsulation [24], and tissue engineering [110]. The hydrogel exists in a liquid sol at room temperature or lower while transforming the state in situ to gelation when the temperature is higher than the critical dissolution temperature, such as the normal body temperature (37 °C) (Figure 3A) [80,111]. Xu, X et al. [112] prepared an injectable and thermosensitive CS, β-sodium glycerophosphate (β-GP), and gelatin hydrogel to achieve continuous release of aspirin and erythropoietin (EPO) to exert anti-inflammatory and tissue regeneration effects, respectively. In their study, CS/β-GP/gelatin hydrogels loaded with aspirin/EPO showed efficacy in anti-inflammation and periodontium regeneration, remodeling the height of the alveolar bone, and providing a great alternative to periodontitis treatments.

As for pH-sensitive hydrogels, they typically have ionizable groups, such as carboxylic acid groups and basic primary amines [113]. The acid–base groups are subjected to variable degrees of ionization, resulting in sensitivity to pH and drug delivery and, ensuring the benefit of the natural control of the inflammatory processes when pH is decreased [114]. Bako, J et al. [115] developed a nanocomposite hydrogel as a pH-sensitive drug delivery system to release MTA and chlorhexidine. While MTA was released from the hydrogel within 12 h, chlorhexidine showed a much longer elution time with strong pH dependence, lasting over 7 days as demonstrated by the bactericidal effect, and it could reduce systemic side effects.

Photosensitive hydrogels induce solvation–gelation changes by exposure to long-range photo light. An important type of photosensitive, engineered gelatin-based material is GelMA, which is well suited for encapsulating PDLCs and has excellent biocompatibility and tunable physical properties [62]. By ultraviolet (UV) irradiation, solutions form irreversibly covalently crosslinked hydrogels in the presence of photoinitiators. However, photo-crosslinking may have disadvantages due to limited UV light penetration and toxic initiators. UV exposure may cause cell/tissue damage, accelerated tissue aging, and even carcinogenesis, and human osteoblasts are less resistant to UV irradiation. In light of these harmful effects, visible light crosslinking has become a popular method for crosslinking hydrogels in recent years [115,116]. Goto, R. et al. [116] evaluated the in vitro feasibility of a visible-wavelength (VW)-light-crosslinked riboflavin (RF)-gelatin-based hydrogel in bone regeneration. The GelMA–RF hydrogel exhibited suitable stiffness for osteoblast differentiation and displayed significantly higher cell viability and gene expression related to osteoblast differentiation than hydrogels photopolymerized with UV light, which means visible-light-crosslinked hydrogels could also be used as a scaffold in bone tissue regeneration.

In addition to the above typical environment-sensitive hydrogels, researchers have also constructed other stimulation-responsive hydrogels or multi-sensitive hydrogels to combine the advantages of multiple single-stimulus-sensitive hydrogels simultaneously. Antimicrobial photodynamic therapy (PDT) is currently used as a novel treatment for periodontitis which generates reactive oxygen species (ROS) for a bactericidal effect [117]. Leung, B. et al. [118] demonstrated that the use of thermosensitive hydrogels containing methylene blue as a topical antimicrobial photodynamic therapy was a promising alternative to treat infectious wounds. Various studies have identified *Porphyromonas gingivalis* as the most pathogenically critical pathogen in developing periodontitis, and gingipain is a key virulence factor [119,120]. Liu, S. et al. [80] designed a gingipain-responsive thermosensitive hydrogel with sustainable release of SDF-1, which effectively controlled the inflammation caused by *P. gingivalis* and enhanced in situ periodontal bone regeneration in vivo (Figure 3B,C). Thus, multi-sensitive hydrogels combine the advantages of various stimulation-responsive hydrogels simultaneously and exhibit significant potential applications in periodontal tissue engineering.

### 3.3. Self-Healing Hydrogels

Self-healing hydrogels refer to a group of hydrogels with the ability to spontaneously repair their structure and function after damage, inspired by the mechanism of self-healing ability in biology [121]. The gelation mechanisms of self-healing hydrogels include dynamic covalent bonds [122], supramolecular bonds [123], and multi-mechanism cross-links.

Self-healing hydrogels have been widely used in wound healing and tissue engineering due to their good self-healing properties [124,125]. Lin, T. K et al. [113] synthesized self-healing hydrogels using Schiff base (also known as imine) linkages between difunctional polyurethane (DFPU) and CS. Depending on the properties of the Schiff bases, these hydrogels are sensitive to low pH and amine-containing molecules and have higher degradation rates in acidic microenvironments and internal porosities, which facilitate the release of drugs or substances(Figure 4). Guo, H et al. [93] reported a double-dynamic network polysaccharide-based hydrogel with rapid gelation, injectability, and excellent self-healing properties as a novel therapy for periodontitis. This hydrogel was synthesized by a dynamic Schiff base formation between -CHO in aldehyde-modified HA and -NH_2_ in glycol CS and a dynamic coordination bond between COO^−^ in aldehyde-modified HA and Fe^3+^, which could transform the sol-gel without external stimuli. The CCK-8 assay showed that this self-healing hydrogel has no cytotoxicity. They further investigated the ability of this hydrogel-loaded with ginsenoside Rg1 and amelogenin to promote periodontal regeneration in periodontitis in vivo. Micro-CT, H&E staining, and immunohistochemical stainings analyses of IL-1, TNF-α, and TGF-β and TRAP indicated that the composite hydrogel could promote alveolar bone regeneration in periodontitis. These injectable self-healing hydrogels appear to have a desirable drug delivery ability and could recover hard tissue destruction in periodontitis.

## 4. Summary and Challenges

This paper reviews the hydrogels applied for periodontal regeneration therapy in recent years, introducing the components and construction strategies for the preparation of relevant hydrogels. Hydrogels have excellent biocompatibility, water retention, and slow release and provide support for cellular interaction and biological function during periodontal regeneration in terms of promoting mesenchymal cell adhesion, migration, proliferation, and differentiation, reducing the inflammatory response, and regulating the immune environment to remodel the structure and function of periodontal tissues. Although remarkable progress has been made in the development of hydrogel therapy in periodontal regeneration, it remains a challenge to provide sufficient mechanical strength and more biological properties to the hydrogels to achieve ideal regenerative effects, something that needs to be taken into account in the future. The periodontal bone ligament-cementum combination is still the major concern of periodontal tissue engineering, especially the recovery of the periodontal ligament. Meanwhile, less research has been conducted on the regeneration of cementum. At present, there are few clinical studies and long-term follow-up reports about the effectiveness of hydrogels for periodontal therapy. Further research is needed to address the relevant issues involved.

## Figures and Tables

**Figure 1 gels-08-00624-f001:**
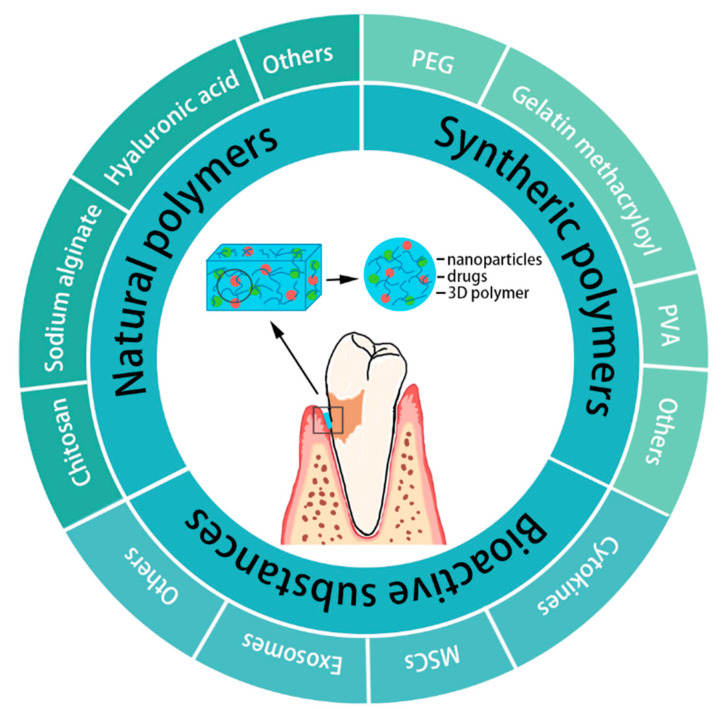
Schematic diagram of the classification of hydrogels in periodontal regeneration.

**Figure 2 gels-08-00624-f002:**
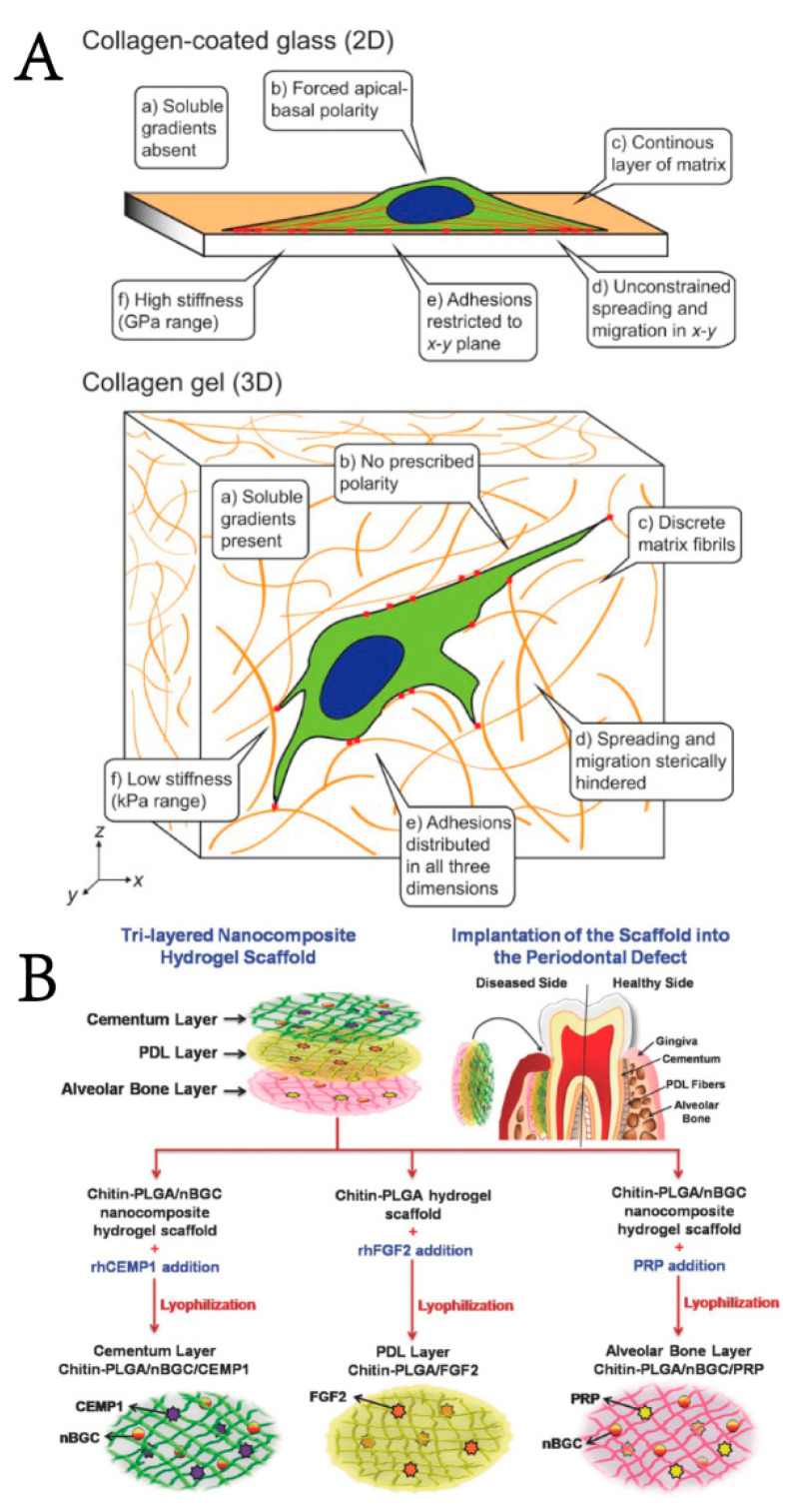
Schematic representation of (**A**) the different bioactive cues encountered by a cell between 2D substrates or in 3D microenvironments (Used with permission of Ref. [102], permission con-veyed through Copyright Clearance Center, Inc, Danvers, MA, USA) and (**B**) a trilayered nanocomposite hydrogel scaffold with a similar structure of cementum-periodontium-alveolar bone for simultaneous and complete periodontal regeneration (ab: alveolar bones; pl: periodontal ligaments; d:dentin; CEMP1: cementum protein-1; FGF-2: fibroblast growth factor-2; PRP: platelet-rich plasma; nBGC: nanobioactive glass ceramic; rhCEMP1: recombinant human cementum protein-1; rhFGF: re-combinant human fibroblast growth factor) (Adapted with permission from [108], Copyright 2017 WILEY‐VCH Verlag GmbH & Co. KGaA, Weinheim, Germany).

**Figure 3 gels-08-00624-f003:**
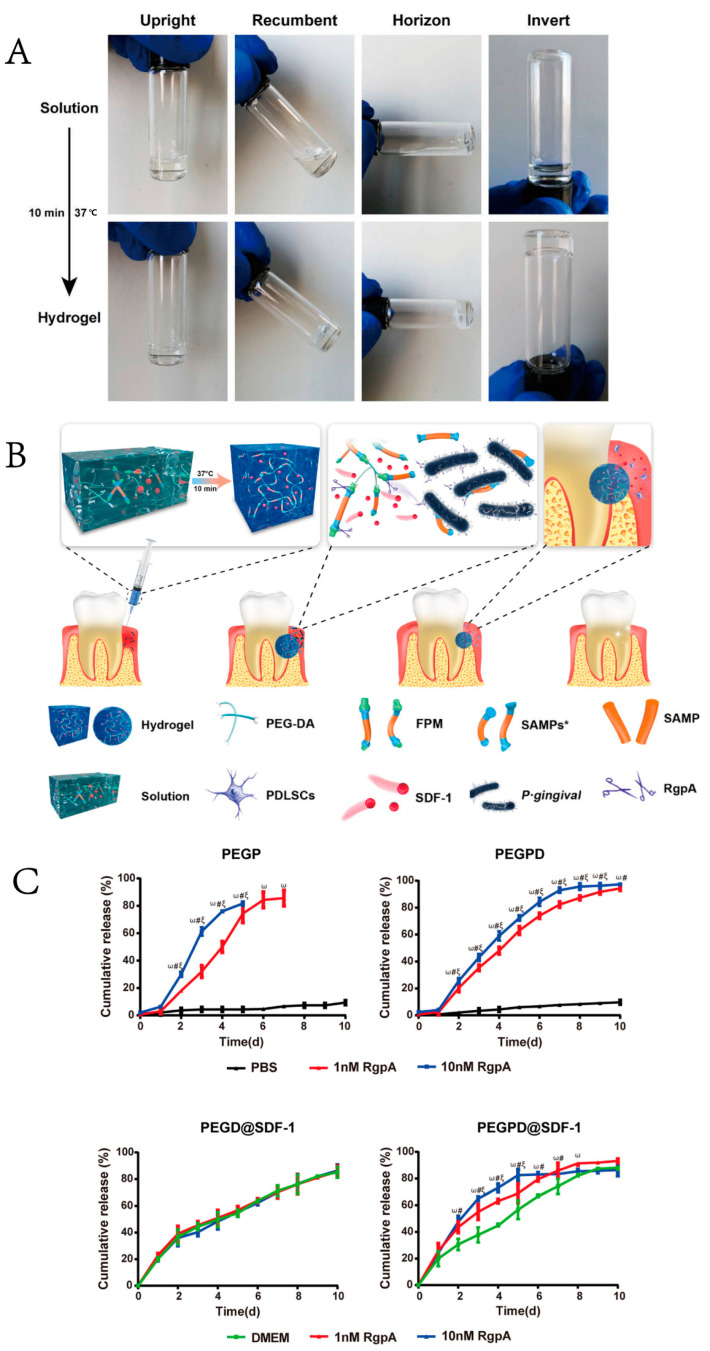
Schematic diagram of the gingipain-responsive thermosensitive hydrogel. (**A**) Photographs of the solution to hydrogel transition by placing the solution in a 37 °C incubator for 10 min. (**B**) Preparation and application of the gingipain-responsive thermosensitive hydrogel. (**C**) Release curves of SAMP (upper) and SDF-1 in different hydrogels. Data are presented as the mean ± SD, *n* = 3. ω, *p* < 0.05 for the PBS/DMEM group vs. the 1 nM RgpA group. #, *p* < 0.05 for PBS/DMEM group vs. 10 nM RgpA group. ξ, *p* < 0.05 for the 1 nM RgpA group vs. the 10 nM RgpA group. (PEG: polyethylene glycol; DA: diacrylate; FPM: functional peptide module; SDF-1: stromal cell derived factor-1; SAMP: short antimicrobial peptide; RgpA: gingipain R1 protein. Adapted with permission from [80], Copyright 2021 American Chemical Society.

**Figure 4 gels-08-00624-f004:**
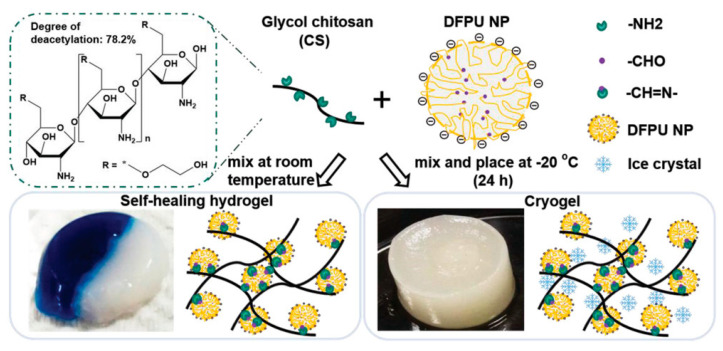
Schematic representation of the simple process to form a self-healing hydrogel or cryogel. (Adapted with permission from Reference [113]).

## Data Availability

Not applicable.

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
