# Peer review of "Advances of Hydrogel Therapy in Periodontal Regeneration—A Materials Perspective Review"

_gels, 2022, doi:10.3390/gels8100624_

Round 1
Reviewer 1 Report
General Comments 1: In this work, the authors tried to evaluate Advances of hydrogel therapy in periodontal regeneration--a materials perspective. The objective of this work has no clear meaning and major important drawbacks of this work should be considered when the decision is made. (1) The English quality is low, and the text is hard to follow. Several badly constructed sentences are found throughout the text, being several grammatical errors easily found. (2) The plan of work is not appropriate, and study was not conducted accurately. (3) Material and method section is incomplete (4) Result and discussion part need more justification and several unrelated sentences were discussed in this section. (5) Presentation of manuscript was poor and most of the part were copied from the thesis of his/her which can be easily identified.
Comments 1: Highlights - Authors should include the Highlights which summaries the research outcomes in 3-5 bullet points not more than 85 characters each. Please include them.
Comment 2: Abstract - The abstract of a research paper should contain a statement of the problem, purpose of the study, methods, data analysis, results, and conclusion. The abstract written in this form acts as an exhaustive statement of results without qualitative explanation and partially explain the significance of the paper, and properly address the accomplished conclusion. Please reshape it.
Comment 3: Graphical abstract -Authors need to include the Graphical abstract that should define the work was carried out and outcome of research. During research what are all the methodology adopted and developed things need to be incorporated in the graphical abstract. Please reshape it.
Comment 4: Introduction should not exceed 3-4 paragraphs (one to two pages maximum). Authors should provide a general introduction about the topic. Also, provide detailed and informative information about published articles. At the end provide the importance of the study and objectives selected for the study.
Comment 5: Objective - The objective of this study has no clear meaning. Authors should include the drawbacks of previous studies. To overcome these drawbacks how authors frame the objective for this study needs to be explained. Please reshape it.
Comment 6: Material method section - Authors should discuss in detail the methodology adopted and procedure followed for conducting this study correlated with previous study. Please reshape it.
Comment 7: Result and discussion part - In this section authors should provide detailed information regarding finding and outcome of research with numerical data and correlate it with previously published studies. The introductory sentences need to be avoided. Please reshape it.
Use the following papers from other researchers.
-Razavi, M., & Khandan, A. (2017). Safety, regulatory issues, long-term biotoxicity, and the processing environment. In Nanobiomaterials Science, Development and Evaluation (pp. 261-279). Woodhead Publishing.
-Farazin, A., Aghadavoudi, F., Motififard, M., Saber-Samandari, S., & Khandan, A. (2021). Nanostructure, molecular dynamics simulation and mechanical performance of PCL membranes reinforced with antibacterial nanoparticles. Journal of Applied and Computational Mechanics, 7(4), 1907-1915.
Reviewer 2 Report
In the present study, Li et al. reviewed the recent advances of hydrogel therapy in periodontal regeneration. The present manuscript introduced the progress of hydrogel materials studied in periodontal tissue and further discussed the construction strategies and up-to-date components of hydrogels. The schemes of the review concerning hydrogel therapy in periodontal regeneration are well-organized. Overall descriptions of previous works and their interpretation seem to be valid and comprehensive.
Some minor revision might be valid for publication.
1. Fig.1 : pollymer—polymer
Line 248: antiosteoclast(?) formation (?) line 249: osteoclast formation (?)
Line 258: transportation therapy (transplantation?)
Line 266: human exfoliated deciduous teeth (stem cells from human exfoliated deciduous teeth?)
Line 270: EV (Extracellular vesicles, EV)
Line 313: conducive(?)
Line 343-347: It would be much comprehensive if authors would describe more in detail why mimicking ECM can be the most attractive advantage of biomimetic hydrogels and why ECM cues are so important for ultimately designing hydrogels to enhance periodontal tissue regeneration.
Fig. 3A: The images of elements are so small that it is hard to identify properly. It is also difficult to see which elements belong to which preparation/application of hydrogels.
Reviewer 3 Report
Title:
Please revise, remove one “-“and add review.
Abstract
Please rewrite” With excellent biocompatibility, water retention, and slow release, 12 hydrogels can simulate the extracellular matrix and provide suitable attachment sites and growth 13 environments for human periodontal cells’”- try not to use excellent
Is this paper a review?
“The purpose of this paper is to review the latest research results”
The figures are high quality.
The discussion is comprehensive.
Please put references in journal style.